# New Clinical Advances in Minimally Invasive Coronary Surgery

**DOI:** 10.3390/jcm14093142

**Published:** 2025-05-01

**Authors:** Shahzad G. Raja

**Affiliations:** Department of Cardiac Surgery, Harefield Hospital, London UB9 6JH, UK; drrajashahzad@hotmail.com; Tel.: +44-1895826511

**Keywords:** endoscopic coronary surgery, hybrid coronary revascularization, minimally invasive coronary surgery, robot-assisted coronary surgery, TECAB

## Abstract

**Background:** Minimally invasive coronary surgery (MICS) has emerged as an alternative approach in the surgical management of coronary artery disease (CAD), offering potential advantages such as reduced surgical trauma, shorter hospital stays, and faster recovery. While conventional coronary artery bypass grafting (CABG) remains the standard treatment for severe CAD, MICS has seen variable adoption due to concerns over procedural complexity, the risk of incomplete revascularization, and the increasing role of percutaneous interventional techniques. **Objectives:** This review examines recent clinical developments in MICS, analyzing its techniques, technological advancements, and the impact on patient outcomes, while also addressing its limitations. **Methods:** This narrative review incorporates studies from PubMed, tracing the evolution of coronary surgery, the refinement of minimally invasive approaches, and the innovations that have enabled the selective implementation of MICS. This review evaluates robot-assisted coronary surgery and totally endoscopic coronary revascularization, discussing their clinical indications and comparative outcomes. **Results:** Advances in imaging, surgical instrumentation, and anesthesia have improved procedural safety and precision, yet MICS remains a selectively utilized technique rather than a universally preferred alternative. Comparative studies demonstrate mixed clinical outcomes, highlighting both the recovery benefits and technical challenges associated with MICS. **Discussion:** Patient selection, preoperative planning, and individualized surgical strategies play a crucial role in optimizing the effectiveness of MICS. Challenges include technical complexity, integration into broader clinical practice, and the need for procedural refinement. While ongoing research continues to address these hurdles, the role of MICS in CAD management remains context-dependent, influenced by case complexity and institutional expertise. **Conclusion:** MICS presents an evolving surgical approach with defined benefits and limitations, requiring careful patient selection and procedural optimization for the best outcomes. This review provides a comprehensive evaluation of recent advances in MICS while acknowledging its challenges and selective application in coronary surgery.

## 1. Introduction

Coronary artery disease (CAD) continues to be a major contributor to morbidity and mortality across the globe [1]. Conventional coronary artery bypass grafting (CABG), which entails heart surgery through sternotomy and the use of cardiopulmonary bypass, has been the primary treatment approach for severe CAD [2]. However, advancements in medical technology have paved the way for minimally invasive coronary surgery (MICS), which offers significant benefits over conventional methods [3]. The goal of MICS is to reduce surgical trauma, limit blood loss and discomfort, expedite recovery with shorter hospital stays, and improve aesthetic outcomes alongside the overall quality of life [4].

Minimally invasive techniques have revolutionized the field of coronary surgery by providing less invasive alternatives to traditional CABG [5]. These techniques minimize the physical and psychological impact on patients, leading to faster recovery times and reduced postoperative complications. MICS offers advantages such as smaller incisions, a decreased requirement for blood transfusions, reduced infection rates, and shorter durations of hospitalization. Additionally, MICS has been associated with improved patient satisfaction and better long-term outcomes [6].

The primary objective of this review is to explore recent clinical advances in minimally invasive coronary surgery. We aim to discuss the latest techniques, technologies, and their impact on patient outcomes. By examining the current state of MICS, we hope to provide insights into its advantages, challenges, and future directions. This review will serve as a comprehensive resource for clinicians, researchers, and healthcare professionals interested in the evolving landscape of coronary surgery.

## 2. Historical Perspective

### 2.1. Evolution of Minimally Invasive Coronary Surgery

Since the first CABG, reported in 1960 [7], surgical techniques have evolved significantly, paving the way for the adoption of minimally invasive approaches. The introduction of MICS marked a paradigm shift in coronary surgery, offering patients safer and less traumatic alternatives to traditional open-heart procedures.

The concept of MICS evolved over decades, with early efforts aimed at reducing surgical invasiveness while maintaining graft patency. One of the foundational contributions to this approach came from Benetti in 1994, who pioneered techniques that minimized surgical trauma while improving patient outcomes [8]. The acronym LAST (left anterior small thoracotomy) was first introduced by Calafiore et al. in 1996, describing a direct left internal mammary artery (LIMA)-to-left anterior descending (LAD) artery anastomosis through a small thoracotomy [9]. Over time, LAST was recognized as an important step in the development of minimally invasive direct coronary artery bypass (MIDCAB), which integrated thoracoscopic LIMA harvesting to refine surgical exposure. Despite initial challenges, such as a steep learning curve, high conversion rates, and surgical complications, MIDCAB with direct LIMA harvesting has emerged as a favorable option for single-vessel LAD disease, offering reduced surgical morbidity and faster recovery compared to conventional coronary artery bypass grafting.

As technology evolved, 1998 witnessed the introduction of the Intuitive Surgical Da Vinci^®^ robot (Sunnyvale, CA, USA) [10]. This breakthrough created new possibilities by utilizing articulating instruments within the chest cavity, minimizing rib spreading, overcoming tremors, and improving accuracy. The first ever robotic CABG using the Da Vinci^®^ robot was performed in Paris in 1998 [11]. Soon, robotic LIMA harvesting combined with MIDCAB became the new gold standard for single-vessel LAD disease in specialized centers [12]. The robotic system made totally endoscopic coronary artery bypass (TECAB) surgery [13], including bilateral IMA harvesting [14], feasible for patients needing multiple grafts. In recent years, TECAB on arrested, as well as beating, hearts has become a reality, with triple-vessel, as well as quadruple-vessel, grafting being performed [15,16,17].

Meanwhile, the multivessel, minimally invasive CABG procedure performed via a mini-thoracotomy has been successfully standardized, yielding outstanding outcomes [17]. Additionally, the use of three-dimensional endoscopes and articulating thoracoscopic instruments has enhanced the procedure in certain centers that operate without robotic systems [18].

### 2.2. Technological Advancements

Advances in technology have been instrumental in shaping the development of MICS [19]. The development of sophisticated surgical tools, enhanced imaging techniques, and innovative procedural approaches has enabled surgeons to carry out intricate coronary interventions with enhanced accuracy and safety [4]. Robotics and endoscopic procedures have further expanded the capabilities of MICS, allowing for more accurate and minimally invasive interventions. These advancements have contributed to improved patient outcomes and the increased adoption of MICS in clinical practice [20].

Founded in 1989, Computer Motion quickly became a pioneer in the surgical robotics industry. Their Automated Endoscope System for Optimal Positioning (AESOP) robotic arm received FDA approval in 1994, marking it as the first telepresence surgical robot. The AESOP system was later upgraded and rebranded as the ZEUS robotic surgical system, which incorporated three remotely operated arms [21].

In 1999, Intuitive Surgical introduced the Da Vinci “Standard” surgical robot for clinical use, which had been initially utilized at the Cleveland Clinic in Ohio the previous year, 1998. While the system faced certain drawbacks, such as the bulky robotic arms and the possibility of collisions, it gained widespread acceptance in clinical settings. Following the 2003 merger between Computer Motion and Intuitive Surgical, the Da Vinci surgical systems became the dominant technology for robot-assisted laparoscopic abdominal procedures, leading to the phasing out of the ZEUS robotic surgical system. Since receiving FDA approval for the Da Vinci Standard system in 2000, three advanced iterations have been released: the S system in 2003, the Si system in 2009, and the Xi system in 2014 [22].

The initial Da Vinci robot, introduced in 2000, was equipped with three arms: one holding an endoscope and two carrying surgical instruments. A four-arm version, approved in 2002, enhanced anatomical visibility and lessened the need for a surgical assistant. The surgeon-controlled console featured two handles designed to reduce hand tremors and scale down movements for improved precision. The Da Vinci S platform, launched in 2006, incorporated a 3D high-definition camera and a touchscreen interface. The 2009 Da Vinci Si model introduced dual console capabilities for the improved training of novice surgeons and included advancements such as an upgraded imaging system and real-time fluorescence imaging. By 2011, the system had been further modified to facilitate single-port surgical access.

The most advanced system to date from Intuitive Surgical is the Da Vinci Xi platform, introduced in 2014 [23,24] (Table 1). The Da Vinci Xi stands out as the world’s most widely adopted multiport robotic surgery system, renowned for its versatility across numerous procedures and specialties. With advanced tools, enhanced visualization, and innovative features, like Firefly fluorescence imaging and integrated table motion, it offers a high level of adaptability. Its standardized design aims to optimize inventory management and improve the overall efficiency in operating rooms. Specifically developed to support procedures across various medical disciplines, the Da Vinci Xi system provides expanded anatomical reach, improved usability, and seamless integration with cutting-edge Da Vinci technologies. This fourth-generation system incorporates 3DHD vision and advanced wristed instruments in a modular and adaptable format. These wristed instruments surpass the range of motion of the human hand, while tremor-filtering and intuitive motion technology enable steady and precise surgical movements. The Da Vinci Xi is also compatible with advanced instruments, such as SureForm staplers (Intuitive Surgical, CA, USA), the Vessel Sealer Extend (Intuitive Surgical, CA, USA), and the Force Bipolar grasper featuring DualGrip technology (Intuitive Surgical, CA, USA) [25].

## 3. Current Techniques in Minimally Invasive Coronary Surgery

At present, minimally invasive coronary surgery techniques are classified according to the method used for mammary artery harvesting [26]. In the early 1990s, Benetti pioneered minimally invasive coronary surgery with the introduction of the left anterior small thoracotomy (LAST) approach [27]. The term “LAST” was first published in 1996 [9]. This technique involved performing a LIMA-to-LAD anastomosis through a fourth intercostal incision to the left of the sternum, with the graft patency evaluated intraoperatively using transthoracic pulsed wave Doppler [9]. The LAST method quickly gained traction in clinical settings. It allowed for IMAs to be harvested either under direct visualization or via an endoscopic approach, with the anastomoses conducted through a mini-thoracotomy. Over time, subsequent publications shifted the focus of LAST to the mini-thoracotomy itself, rather than the specific grafting method employed [28]. Shortly after the initial publication of LAST in 1996, the term “MIDCAB” emerged [29]. The term MIDCAB quickly gained popularity in clinical practice as a designation for any technique involving minimally invasive coronary revascularization. In current clinical practice, minimally invasive coronary surgery is performed under direct vision, endoscope-assisted, robot-assisted, or totally endoscopic.

### 3.1. Direct Vision

In this approach, the IMA is harvested under direct vision through the same incision that is utilized for the coronary bypass surgery. When available, a specialized retractor is utilized for IMA harvesting. The location of the mini-thoracotomy is determined by the patient’s individual anatomy and the surgeon’s preferences. Generally, an incision measuring 4–6 cm is created at the fourth or fifth intercostal space along the mid-axillary line. In the µCAB technique, the use of a specially designed retractor for subxiphoid access during IMA harvesting has been reported [30]. Another variation involves a lower distal mini-sternotomy (TM-OPCAB) to facilitate access to more distal coronary targets [31].

Anastomoses may be performed either on a beating heart or with the heart arrested. When performed on a beating heart, cardiac stabilizers are employed to secure the target vessels during the procedure [32]. Anaortic or no-touch aorta techniques for off-pump CABG are linked to a reduction in neurological complications and are described as minimally invasive off-pump anaortic coronary artery bypass (MACAB) [32,33]. For grafting lateral and inferior regions using off-pump methods, cardiac stabilizers, such as those referenced by Kikuchi et al. in minimally invasive coronary artery bypass grafting (MICS CABG), are essential [34]. During grafting on an arrested heart through a mini-thoracotomy, cardioplegia is administered following the application of an aortic cross-clamp. This technique has been referred to by various terms, including endo-CABG (endoscopic coronary artery bypass grafting) [35] and total coronary revascularization via a left anterior thoracotomy (TCRAT) [36].

### 3.2. Endoscope-Assisted Coronary Artery Bypass

The endoscope-assisted harvesting of IMAs and grafting through a mini-thoracotomy have been utilized for both single-vessel and multivessel coronary revascularization. Both venous and arterial conduits have been effectively employed in this approach. Among the arterial conduits, the IMA, gastroepiploic artery, and radial arteries have been harvested endoscopically [37].

Endoscopic atraumatic coronary artery bypass (Endo-ACAB) is a technique where the IMA is harvested endoscopically via ports placed near the third, fifth, and seventh intercostal spaces, located between the mid and anterior axillary lines. This approach reduces rib spreading compared to direct-vision retractors [38]. Single-lung ventilation is initiated, along with carbon dioxide insufflation, to enable the harvesting of one or both IMAs. After the IMA is harvested, the coronary target is located, and the mini-thoracotomy is performed [39]. To ensure accuracy, the mini-thoracotomy site can be marked with a needle [40].

Grafts are then performed either on a beating heart or an arrested heart through the mini-thoracotomy. In cases where the grafts are performed on a beating heart, cardiac stabilizers ensure precision during the procedure. For grafting on an arrested heart, peripheral cardiopulmonary bypass is established before proceeding with the anastomoses. This versatile technique has been adopted for both off-pump and on-pump approaches, depending on patient-specific requirements [35].

### 3.3. Robot-Assisted Coronary Artery Bypass

The introduction of port-access internal mammary artery (IMA) harvesting enabled the adaptation of IMA harvest instruments for use with robotic surgical systems. Robot-assisted, minimally invasive coronary artery bypass grafting was first implemented in the late 1990s [11,12,41]. This technique utilizes a surgical robot to harvest the IMA through endoscopic ports, significantly reducing chest wall trauma. Once the IMA has been harvested, the anastomoses are completed via a mini-thoracotomy [42,43]. Procedures of this type are commonly referred to by terms such as MIDCAB or robot-assisted, minimally invasive direct coronary artery bypass (RAMIDCAB).

### 3.4. Totally Endoscopic Coronary Artery Bypass

Completely closed-chested, robot-assisted revascularization represents the least invasive method among current minimally invasive surgical techniques. Initial studies on animals were conducted to evaluate whether both IMA harvesting and the anastomosis could be achieved entirely endoscopically, a process referred to as endoscopic coronary artery bypass grafting (ECABG) [44]. In totally endoscopic coronary artery bypass (TECAB), both the IMA harvest and anastomosis are performed using a fully endoscopic approach. This technique accommodates the use of either on-pump or off-pump methods to address single or multivessel coronary disease. In TECAB techniques, supplementary parasternal and subcostal ports are utilized to facilitate robotic anastomosis. Nonetheless, the scarcity of cardiac stabilizers compatible with robotic systems presents obstacles to advancing off-pump, fully endoscopic robotic surgeries [45].

## 4. Advances in Surgical Techniques and Technologies

### 4.1. Enhanced Imaging Techniques

Enhanced imaging techniques have significantly advanced MICS, offering unprecedented precision and clarity in visualizing coronary anatomy. Intraoperative imaging modalities, such as three-dimensional (3D) imaging and fluoroscopy, have become indispensable tools for surgeons, enabling accurate navigation and graft placement during procedures. These technologies not only enhance the surgeon’s ability to identify and address complex anatomical challenges but also reduce the likelihood of intraoperative errors. For instance, 3D imaging provides a detailed spatial understanding of the coronary vasculature, while fluoroscopy offers real-time guidance during grafting, ensuring optimal outcomes. Recent advancements in imaging software have further improved the integration of these modalities with robotic systems, streamlining surgical workflows and enhancing procedural efficiency [5].

Preoperative imaging and simulation technologies have also played a pivotal role in advancing MICS. High-resolution computed tomography (CT) and cardiac magnetic resonance imaging (CMR) allow for comprehensive preoperative assessments, enabling surgeons to develop tailored surgical strategies based on patient-specific anatomy. These imaging techniques facilitate the identification of potential complications and the optimization of surgical plans, thereby improving the safety and efficacy of MICS procedures. The integration of artificial intelligence (AI) in cardiovascular imaging has significantly improved the accuracy of preoperative planning by automating image analysis and risk stratification [46]. Additionally, innovations in plaque characterization using CT have provided valuable insights into disease severity and progression, aiding in the selection of appropriate surgical interventions [47].

### 4.2. Advanced Surgical Tools

The development of specialized instruments for MICS has significantly improved the precision and safety of coronary interventions. Robotic systems, endoscopic tools, and automated devices have expanded the capabilities of MICS, allowing for more accurate and minimally invasive procedures. These advanced surgical tools have contributed to improved patient outcomes, reduced surgical trauma, and faster recovery times.

Robotic systems have revolutionized MICS by enhancing precision, dexterity, and visualization. These systems, such as the Da Vinci Xi, allow surgeons to perform complex procedures with greater accuracy through small incisions, minimizing trauma to surrounding tissues. Robotic platforms integrate advanced imaging technologies, including 3D high-definition visualization, which provides a detailed view of the surgical field. Recent studies highlight the role of robotic systems in improving surgical outcomes, reducing operative times, and minimizing complications [48]. Additionally, robotic systems have facilitated the development of TECAB, enabling complete revascularization without the need for sternotomy. These advancements underscore the transformative impact of robotics on the field of MICS.

The upcoming Da Vinci 5 robotic system is designed to enhance surgical capabilities and improve patient outcomes by incorporating advanced technologies. These advancements include Force Feedback technology, an upgraded Da Vinci vision system, and improved ergonomic features to support surgeons during procedures. The Intuitive Surgical Endoscopic Instrument Control System (Model IS5000) enables the precise handling of a wide range of endoscopic instruments. These instruments include rigid endoscopes, endoscopic dissectors for both blunt and sharp tasks, scissors, scalpels, forceps, needle holders, retractors, and electrocautery tools. Additionally, the system supports various procedures such as tissue manipulation through grasping, cutting, dissection, approximation, ligation, suturing, and electrocautery. It is also equipped to handle the delivery and placement of specialized probes, including microwave and cryogenic ablation devices, making it a versatile tool for modern surgical needs [49]. 

Endoscopic tools have become indispensable in MICS, offering surgeons enhanced access and visualization of the surgical site. These instruments are designed to navigate through small incisions, reducing the need for extensive dissection and minimizing patient trauma. Innovations in endoscopic technology, such as articulating instruments and high-definition cameras, have further improved the precision and safety of procedures. The endoscopic tools facilitate procedures like IMA harvesting and anastomosis [4]. These tools have also been integrated with robotic systems, creating a synergistic effect that enhances the capabilities of minimally invasive techniques. The continued evolution of endoscopic tools is expected to drive further advancements in MICS.

Automated devices have emerged as a critical component of MICS, streamlining procedures and improving efficiency. These devices include automated suturing systems, vessel-sealing technologies, and advanced cardiac stabilizers, all of which contribute to the precision and safety of surgical interventions. Recent developments in AI have further enhanced the functionality of automated devices, enabling real-time adjustments and decision-making during surgery. A 2024 study discusses the integration of AI-driven automated devices in robot-assisted surgery, highlighting their potential to reduce human error and improve patient outcomes [50]. These innovations not only enhance the surgeon’s capabilities but also pave the way for more consistent and reproducible results in minimally invasive coronary procedures.

### 4.3. Innovations in Anesthesia and Pain Management

Advancements in anesthesia and pain management have significantly improved the recovery experience for patients undergoing MICS. Regional anesthesia techniques, such as erector spinae plane blocks and paravertebral blocks, have gained prominence for their ability to provide targeted pain relief while minimizing systemic side effects. These methods lessen dependence on opioids, which can lead to undesirable side effects like respiratory depression and prolonged recovery times. Incorporating erector spinae plane blocks in MICS significantly lowers postoperative pain levels and reduces the duration of mechanical ventilation compared to conventional techniques [51]. These findings underscore the importance of regional anesthesia in enhancing patient comfort and facilitating faster recovery.

Enhanced recovery after surgery (ERAS) protocols have also been instrumental in optimizing pain management for MICS patients. These protocols emphasize multimodal pain management strategies that combine regional anesthesia, non-opioid analgesics, and minimally invasive techniques to achieve superior outcomes. The role of ERAS protocols in reducing hospital stays and improving patient satisfaction by promoting early mobilization and minimizing opioid consumption is well established [52]. The integration of ERAS principles into MICS has not only improved perioperative outcomes but also set new standards for patient-centered care in cardiac surgery [53].

In addition to regional anesthesia and ERAS protocols, technological innovations have contributed to advancements in pain management. For example, the use of ultrasound-guided nerve blocks has enhanced the precision and efficacy of regional anesthesia techniques. Furthermore, the development of long-acting local anesthetics and continuous infusion systems has provided sustained pain relief, reducing the need for systemic medications. The application of these technologies in MICS demonstrates their potential to improve both short-term and long-term recovery outcomes [20]. These innovations represent a paradigm shift in the management of perioperative pain, ensuring that patients undergoing MICS experience a smoother and more comfortable recovery process.

## 5. Patient Selection and Preoperative Considerations

Patient selection plays a pivotal role in determining the success of MICS. The suitability of a patient for MICS is affected by numerous determinants, including the severity and complexity of CAD, anatomical considerations such as vessel accessibility, and the patient’s overall health and comorbidities. For instance, patients with isolated LAD artery disease are often ideal candidates for MIDCAB, while those with multivessel disease may require more comprehensive evaluations to determine the feasibility of robot-assisted or totally endoscopic approaches. Anatomical imaging in preoperative planning plays a crucial role in identifying patients who are most likely to benefit from MICS while minimizing the procedural risks [5].

Risk assessment and stratification are essential components of the patient selection process. Risk calculators, such as the Society of Thoracic Surgeons (STS) risk score and the EuroSCORE II, are commonly used to evaluate the surgical risk and predict outcomes. These scoring systems help identify high-risk patients who may benefit from the reduced invasiveness of MICS compared to traditional open-heart surgery. Frailty assessments and functional status evaluations play a crucial role in refining patient selection criteria for MICS. These evaluations help ensure that patients with limited physiological reserves are not exposed to unnecessary surgical risks, thereby enhancing overall outcomes [54].

Multidisciplinary team (MDT) evaluations are critical in ensuring appropriate patient selection for MICS. These teams typically include cardiologists, cardiac surgeons, anesthesiologists, and imaging specialists who collaborate to develop personalized treatment plans. MDT discussions incorporate clinical guidelines, imaging findings, and patient preferences to determine the most suitable surgical approach. It is being increasingly recognized that MDT evaluations are essential for achieving optimal outcomes, especially in complex cases involving multivessel disease or prior surgical interventions [55]. By integrating diverse expertise, MDTs enhance decision-making and ensure that MICS is offered to patients who are most likely to benefit from its minimally invasive nature.

## 6. Short-Term and Long-Term Outcomes

When compared to conventional CABG, MICS offers significant benefits in both immediate and extended outcomes. In the short term, it minimizes surgical trauma, resulting in quicker recovery, a reduced length of hospital stay, and lower rates of postoperative complications, such as infections and bleeding. Additionally, patients undergoing MICS often experience shorter durations of mechanical ventilation and a reduced need for transfusions, highlighting its effectiveness in enhancing perioperative recovery [56].

The long-term outcomes of MICS are equally promising, with studies demonstrating comparable or superior graft patency and survival rates relative to traditional CABG. The use of advanced imaging and robot-assisted techniques in MICS has contributed to precise graft placement, which is critical for long-term success. MICS has been shown to deliver durable outcomes, particularly for patients with isolated LAD artery disease, where graft patency rates remain consistently high over long-term follow-up periods [57]. Additionally, the minimally invasive nature of MICS has been associated with an improved quality of life and patient satisfaction, as it minimizes the physical and psychological stress associated with larger incisions and prolonged recovery times. Table 2 summarizes key studies published in the last 5 years comparing MICS with conventional CABG.

## 7. Challenges and Future Directions

### 7.1. Technical Challenges

In recent years, MICS has advanced significantly, though numerous technical hurdles remain. One of the primary obstacles is the steep learning curve for surgeons. Mastering MICS techniques demands extensive training and experience, as these procedures often require precise coordination and familiarity with advanced tools. This challenge is compounded by the limited availability of specialized training programs and mentorship opportunities [4,64]. Additionally, multivessel revascularization with MICS remains a challenge, as conventional techniques often allow for more complete revascularization without the procedural complexities associated with robot- or thoracoscope-assisted approaches. Given these technical constraints, patient selection plays a crucial role in ensuring optimal surgical outcomes, and some cases may necessitate a hybrid approach to balance the minimally invasive benefits with complete revascularization needs.

Technological limitations also play a significant role. Current robotic systems and endoscopic tools, while innovative, still have room for improvement in terms of their precision, reliability, and cost-effectiveness. For instance, the lack of tactile feedback in robotic systems can make certain procedures more challenging for surgeons, affecting the graft placement accuracy [65]. Additionally, the high costs associated with acquiring and maintaining these technologies can be prohibitive for many healthcare institutions, particularly in resource-limited settings [66]. Regulatory constraints further restrict the widespread adoption of robotics in coronary surgery, as robot-assisted procedures are often limited to internal mammary artery harvesting, rather than full revascularization, in certain regions. This regulatory hurdle contributes to cost inefficiencies, where robotic assistance may offer marginal benefits while significantly increasing the financial burden for healthcare systems.

The complexity of certain coronary lesions further complicates the application of MICS. Patients with comorbidities or unique anatomical variations often present challenges that require tailored approaches, which may not always be feasible with the existing MICS techniques. The risk of incomplete surgical revascularization is a well-recognized limitation of current MICS approaches, particularly in multivessel disease management. When exploring the outcomes of non-robotic MICS techniques, it is evident that some direct-vision or endoscope-assisted approaches may offer comparable or even superior results while avoiding the extended operative times and postoperative complications associated with robot-assisted surgery. Additionally, the duration of MICS procedures is often significantly longer, leading to increased time on mechanical ventilation, prolonged ICU stays, and higher risks of respiratory complications—factors that impact hospital resource utilization and cost-effectiveness.

Addressing these challenges requires a multifaceted approach. Ongoing research is essential to develop more advanced and user-friendly technologies, such as robotic systems with enhanced capabilities, improved imaging tools, and tactile feedback features. Training programs need to be expanded and standardized, ensuring that surgeons gain the necessary expertise without excessive procedural inefficiencies. Furthermore, fostering collaboration between clinicians, researchers, and industry partners is crucial for driving innovation and overcoming these obstacles. A more balanced approach to integrating MICS into routine clinical practice is needed, ensuring that its application is reserved for cases where minimally invasive techniques provide clear advantages without compromising surgical completeness.

### 7.2. Research and Development

Research and development (R&D) are the backbone of advancing MICS. Clinical trials play a crucial role in evaluating new techniques and technologies, such as robot-assisted surgery and endoscopic tools, which aim to improve precision, reduce invasiveness, and enhance recovery outcomes [4]. Recent efforts have focused on refining patient selection criteria by leveraging advanced diagnostic tools to tailor surgical approaches to individual needs [67]. Additionally, innovative surgical methods like MIDCAB are being studied for their potential to become standard practices for specific patient groups. Alongside these innovations, research into strategies to mitigate postoperative complications, such as thromboembolism and neurological risks, remains a priority, with enhanced recovery protocols gaining attention [4].

Collaborative efforts between academic institutions, healthcare organizations, and industry partners are central to driving progress in MICS. These partnerships facilitate the development of advanced robotic systems, imaging tools, and surgical techniques while addressing the financial and logistical challenges associated with these technologies [4]. Expanding and standardizing training programs also play a critical role in equipping surgeons with the necessary skills to adopt these methods effectively. With ongoing investment in R&D and a commitment to interdisciplinary collaboration, the field of MICS is poised to overcome the current limitations, ultimately improving patient outcomes and making these procedures more accessible on a global scale.

### 7.3. Integration into Clinical Practice

The integration of MICS into clinical practice is a multifaceted process that hinges on the adoption and dissemination of advanced techniques. Training and education for healthcare professionals are paramount to ensuring the safe and effective implementation of MICS. Comprehensive training programs, including simulation-based learning and mentorship opportunities, have been shown to accelerate the learning curve for surgeons and improve procedural outcomes [19]. Additionally, the establishment of standardized protocols and clinical guidelines is critical for maintaining consistency and safety across institutions [4].

Collaboration between healthcare providers, policymakers, and industry partners plays a pivotal role in facilitating the widespread adoption of MICS. Such partnerships enable the development of innovative technologies, the sharing of best practices, and the allocation of resources to support training and infrastructure [4]. Furthermore, interdisciplinary collaboration fosters the creation of tailored solutions to address specific challenges, such as cost barriers and access disparities [19]. By prioritizing these efforts, the integration of MICS into routine clinical practice can lead to improved patient outcomes and broader accessibility to minimally invasive procedures.

## 8. Conclusions

Recent advancements in MICS have revolutionized the treatment of CAD through the refinement of techniques such as MIDCAB, robot-assisted coronary artery bypass, and TECAB. Innovations in imaging, surgical tools, and anesthesia protocols have enhanced precision and safety, resulting in reduced trauma, complications, and recovery times, ultimately improving patient satisfaction and quality of life. As research efforts focus on enhancing techniques, advancing tools, and utilizing AI-powered personalized medicine, the field progresses steadily. Collaborative efforts among clinicians, researchers, and industry partners are essential for integrating these advancements into clinical practice, ensuring better outcomes, and expanding global accessibility.

## Figures and Tables

**Table 1 jcm-14-03142-t001:** The four generations of the Da Vinci robotic system.

Generation	Year Introduced	Key Features
Da Vinci Standard	2000	Three arms (one endoscope, two instruments), console with two handles, minimized hand tremors, scaled-down movements for precision
Da Vinci S System	2003	3D HD camera, touchscreen display, improved ergonomics, enhanced vision system
Da Vinci Si System	2009	Dual console surgery, improved training for non-expert surgeons, upgraded image system, real-time fluorescence imaging, enhanced 3D HD vision, better instrument control
Da Vinci Xi System	2014	Multiport access, advanced features for improved anatomical exposure, reduced reliance on surgical assistant, modular and flexible design, improved energy efficiency, enhanced vision and instrument capabilities

**Table 2 jcm-14-03142-t002:** Key studies published between 2020 and 2025 comparing MICS with conventional CABG.

Author	Year of Publication	Total Number of Patients	Number of MICS Patients	Number of CCABG Patients	Key Outcomes
Ushioda et al. [58] *	2024	1220	149	149	Similar hospital stays, intensive care unit stays, postoperative complications, MACCE, and 30-day mortality; lower total graft number and fewer distal anastomoses with MICS.
Guangxin et al. [59]	2024	104	52	52	Comparable graft patency rates; reduced blood loss and wound complications with MICS.
Huang et al. [60]	2023	444	179	265	Similar number of grafts, perioperative complications, and mortality; reduced blood loss, shorter hospital stays, and longer operation durations with MICS
Liang et al. [61] *	2022	344	172	172	Similar in-hospital outcomes; shorter hospital stays and faster recovery with MICS.
Xu et al. [62] *	2020	536	85	85	Similar in-hospital outcomes; shorter hospital stays and faster recovery with MICS.
Stanislawski et al. [63] *^,†^	2020	194	111	93	Similar in-hospital outcomes; lower chest tube drainage and shorter hospital stay with MICS.

* Propensity-matched study. ^†^ Comparison of MIDCAB with full sternotomy off-pump LIMA-LAD. CCABG = conventional coronary artery bypass grafting; MACCE = major adverse cardiovascular and cerebrovascular event; MICS = minimally invasive coronary surgery.

## Data Availability

No new data were created or analyzed in this study. Data sharing is not applicable to this article.

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
