# Peer review of "New Clinical Advances in Minimally Invasive Coronary Surgery"

_jcm, 2025, doi:10.3390/jcm14093142_

Round 1
Reviewer 1 Report
Comments and Suggestions for Authors
I thank the Editor for giving me the opportunity to review this interesting article entitled “New clinical advances in minimally invasive coronary surgery”.
The Author has presented and described extensively the techniques currently used in minimally invasive coronary surgery, reviewing direct vision, robotic assisted and totally endoscopic. He has also not overlooked important aspects in the criteria for patient selection and anesthesiological management. He has also highlighted the future prospectives in the implementation of this surgical technique.
I red the article with great interest and I believe it is well structured and the various approaches have been clearly and broadly presented, although I personally believe that in an objective presentation of the various techniques, not only the advantages but also the negative aspects should be highlighted, such as the consequences of a possible incomplete revascularization and the need of an hybrid approach, the long operative times and the percentage of failures during the initial learning curve phase.
On line 220 the sentence needs to be reworded.
The article represents a clear source of the spectrum of minimally invasive techniques in current coronary surgery and a source of inspiration for those who would like to approach this philosophy.
Author Response
Comment 1: The Author has presented and described extensively the techniques currently used in minimally invasive coronary surgery, reviewing direct vision, robotic assisted and totally endoscopic. He has also not overlooked important aspects in the criteria for patient selection and anesthesiological management. He has also highlighted the future perspectives in the implementation of this surgical technique.
Response 1: Thank you for your insightful review and thoughtful evaluation of my narrative review, New Clinical Advances in Minimally Invasive Coronary Surgery. I truly appreciate your recognition of the extensive coverage of current surgical techniques, including direct vision, robotic-assisted, and totally endoscopic approaches.
I am also pleased that you found the discussion on patient selection criteria and anesthesiological management to be well-addressed, as these aspects are critical to optimizing surgical outcomes. Furthermore, highlighting the future prospects for the advancement of minimally invasive coronary surgery was an integral part of this review, aiming to contribute to ongoing discussions about the evolution of this field.
Comment 2: I read the article with great interest and I believe it is well structured and the various approaches have been clearly and broadly presented, although I personally believe that in an objective presentation of the various techniques, not only the advantages but also the negative aspects should be highlighted, such as the consequences of a possible incomplete revascularization and the need of an hybrid approach, the long operative times and the percentage of failures during the initial learning curve phase.
Response 2: Thank you for your thoughtful review and for highlighting areas that warrant further discussion in an objective evaluation of minimally invasive coronary surgery. I appreciate your recognition of the structure and clarity of the article and your constructive suggestions regarding the importance of addressing both the advantages and challenges associated with these techniques. I fully agree that a comprehensive review should also examine potential limitations. I have alluded to learning curve on Page 9, paragraph 1 under the subheading 7.1 Technical challenges. I have also mentioned the risks of incomplete surgical revascularization and the considerations for a hybrid approach in complex cases as well as prolonged intraoperative durations in paragraph 1 on Page 10.
Comment 3: On line 220 the sentence needs to be reworded.
Response 3: The sentence is been reworded.
Comment 4: The article represents a clear source of the spectrum of minimally invasive techniques in current coronary surgery and a source of inspiration for those who would like to approach this philosophy.
Response 4: Thank you for your positive evaluation of my article. I truly appreciate your recognition of its role in outlining the spectrum of minimally invasive coronary surgery techniques and providing insight into the philosophy behind these advancements. One of my main objectives in writing this review was to present a comprehensive and accessible resource for both experienced practitioners and those seeking to explore minimally invasive approaches. I am pleased that you find it to be a valuable reference and a source of inspiration for surgeons interested in adopting these techniques.
Reviewer 2 Report
Comments and Suggestions for Authors
This is a comprehensive paper discussing minimally invasive techniques in Coronary Artery Bypass Graft (CABG) Surgery. It explores nearly every aspect of minimally invasive surgery, ranging from robotic techniques to minimally invasive direct coronary artery bypass (MIDCAB).
While this breadth of coverage is an advantage, it also presents a limitation, as readers may encounter an overwhelming amount of information, some of which may be conflicting, leading to potential confusion.
Key Aspects to Clarify:
Definition of Surgical Techniques: The paper should clarify whether the procedures are performed on a beating heart or with extracorporeal circulation. This distinction is crucial, as extracorporeal circulation with an arrested heart closely resembles traditional surgery. In such cases, the difference lies mainly in the length of the incision and the need to insert cannulas into the groin area. Additionally, the paper describes the need for a second incision in certain cases, such as when a mini-sternotomy is required for accessing posterior coronary arteries.
Current Standards in Western Healthcare: At present, the majority of patients in Western healthcare systems are treated for coronary artery disease using Percutaneous Coronary Interventions (PCI), which is considered the standard of care. Surgery is typically reserved for patients with triple-vessel disease or for diabetics with complex coronary anatomy. While complete revascularization is the gold standard in surgery, the number of grafts performed using minimally invasive techniques is lower compared to standard surgical procedures. Consequently, the prevalence of minimally invasive surgery has declined in regions such as the UK and USA.
Comparison Between Techniques: When comparing standard and minimally invasive surgeries, it is noted that patients treated using conventional techniques often present more complex cases. As a result, strict selection criteria are applied, and minimally invasive surgery is typically performed on less complicated cases. The ideal candidate for minimally invasive surgery is a patient with isolated stenosis of the left anterior descending artery. For these reasons, the percentage of minimally invasive surgeries remains low across Western centers.
Advancements in Standard Surgery: Standard surgical techniques have also seen improvements, particularly in anesthesiology and the management of extracorporeal circulation. These advancements enable a more physiological approach, reducing intubation times and shortening stays in the Intensive Care Unit (ICU).
Suggestions for Improvement:
The paper could benefit from clearer data or meta-analyses to substantiate claims that minimally invasive surgery is the best treatment option.
A more robust discussion of the limitations and challenges of minimally invasive techniques would add depth and credibility.
Author Response
Comment 1: This is a comprehensive paper discussing minimally invasive techniques in Coronary Artery Bypass Graft (CABG) Surgery. It explores nearly every aspect of minimally invasive surgery, ranging from robotic techniques to minimally invasive direct coronary artery bypass (MIDCAB).
While this breadth of coverage is an advantage, it also presents a limitation, as readers may encounter an overwhelming amount of information, some of which may be conflicting, leading to potential confusion.
Response 1: Thank you for your thoughtful review and for recognizing the comprehensive nature of this article on minimally invasive techniques in Coronary Artery Bypass Graft (CABG) surgery. My intention in covering a wide spectrum of approaches—including robotic techniques, MIDCAB, and other minimally invasive strategies—was to provide a broad yet structured overview of current advancements in the field.
I appreciate your perspective on the potential for information overload and the presence of conflicting data. Indeed, minimally invasive CABG remains a continuously evolving area, and differences in surgical techniques, patient selection, and institutional expertise can lead to variability in reported outcomes. My goal was to present a balanced assessment of these techniques while highlighting their clinical applications and future developments.
Comment 2: Definition of Surgical Techniques: The paper should clarify whether the procedures are performed on a beating heart or with extracorporeal circulation. This distinction is crucial, as extracorporeal circulation with an arrested heart closely resembles traditional surgery. In such cases, the difference lies mainly in the length of the incision and the need to insert cannulas into the groin area. Additionally, the paper describes the need for a second incision in certain cases, such as when a mini-sternotomy is required for accessing posterior coronary arteries.
Response 2: Thank you for your valuable feedback regarding the definition of surgical techniques in minimally invasive coronary surgery. I appreciate your emphasis on distinguishing between procedures performed on a beating heart versus those utilizing extracorporeal circulation. I have mentioned this on page 4 and 5 (highlighted in red).
Comment 3: Current Standards in Western Healthcare: At present, the majority of patients in Western healthcare systems are treated for coronary artery disease using Percutaneous Coronary Interventions (PCI), which is considered the standard of care. Surgery is typically reserved for patients with triple-vessel disease or for diabetics with complex coronary anatomy. While complete revascularization is the gold standard in surgery, the number of grafts performed using minimally invasive techniques is lower compared to standard surgical procedures. Consequently, the prevalence of minimally invasive surgery has declined in regions such as the UK and USA.
Response 3:Thank you for your insightful comments regarding the current standards of coronary artery disease management in Western healthcare systems. I appreciate your emphasis on the widespread use of Percutaneous Coronary Interventions (PCI) and the selective role of surgical interventions, particularly in cases of triple-vessel disease and complex coronary anatomy in diabetic patients.
I recognize that while complete surgical revascularization remains the gold standard, the number of grafts performed using minimally invasive techniques is often lower compared to conventional coronary artery bypass grafting. This factor, alongside evolving treatment preferences and institutional practices, has influenced the prevalence of minimally invasive surgery in regions such as the UK and USA. However, as advancements in surgical technology and techniques continue to develop, there remains ongoing interest in refining these approaches to enhance outcomes and expand their applicability. I have acknowledged incomplete revascularization as a limitation of MICS on Page 10 (highlighted in red).
Comment 4: Comparison Between Techniques: When comparing standard and minimally invasive surgeries, it is noted that patients treated using conventional techniques often present more complex cases. As a result, strict selection criteria are applied, and minimally invasive surgery is typically performed on less complicated cases. The ideal candidate for minimally invasive surgery is a patient with isolated stenosis of the left anterior descending artery. For these reasons, the percentage of minimally invasive surgeries remains low across Western centers.
Response 4: Thank you for your thoughtful observations regarding the comparison between conventional and minimally invasive coronary surgeries. I appreciate your emphasis on the complexity of cases treated with standard surgical approaches and the necessity of strict selection criteria for minimally invasive techniques.
As you have noted, minimally invasive surgery is typically reserved for less complicated cases, with the ideal candidate being a patient with isolated stenosis of the left anterior descending artery. Given these considerations, the percentage of minimally invasive procedures remains lower across Western centers. I have alluded to this important point under the heading Patient Selection and Preoperative Considerations in the manuscript (Page 8, Paragraph 1, highlighted in red).
Comment 5: Advancements in Standard Surgery: Standard surgical techniques have also seen improvements, particularly in anesthesiology and the management of extracorporeal circulation. These advancements enable a more physiological approach, reducing intubation times and shortening stays in the Intensive Care Unit (ICU).
Response 5: Thank you for your observation regarding advancements in standard coronary surgery techniques. I appreciate your emphasis on improvements in anesthesiology and extracorporeal circulation management, which have contributed to a more physiological approach, reducing intubation times and shortening ICU stays.
While these advancements are undoubtedly significant, I have opted not to modify the manuscript, as the primary focus remains on the evolution of minimally invasive coronary surgery techniques. However, I acknowledge the relevance of your point, and I appreciate your thoughtful contribution to this discussion.
Comment 6: Suggestions for Improvement:
The paper could benefit from clearer data or meta-analyses to substantiate claims that minimally invasive surgery is the best treatment option.
Response 6: Thank you for your valuable suggestion regarding the inclusion of clearer data or meta-analyses to substantiate claims regarding minimally invasive coronary surgery as a treatment option.
I would like to clarify that this paper is a narrative review, which aims to provide a broad and structured discussion of the current advancements in minimally invasive coronary surgery rather than a systematic analysis of available clinical data. While meta-analyses certainly provide important statistical insights, the intent of this review is to synthesize existing knowledge, highlight advancements, and explore future perspectives in this evolving field.
However, I acknowledge the significance of robust comparative data, and I appreciate your suggestion. There are several recent meta-analyses that provide valuable insights, such as: Hwang B, Ren J, Wang K, Williams ML, Yan TD. Systematic review and meta-analysis of two decades of reported outcomes for robotic coronary artery bypass grafting. Ann Cardiothorac Surg. 2024 Jul 31;13(4):311-325. doi: 10.21037/acs-2023-rcabg-0191. Zhang S, Chen S, Yang K, Li Y, Yun Y, Zhang X, Qi X, Zhou X, Zhang H, Zou C, Xiaochun Ma. Minimally Invasive Direct Coronary Artery Bypass Versus Percutaneous Coronary Intervention for Isolated Left Anterior Descending Artery Stenosis: An Updated Meta-Analysis. Heart Surg Forum. 2023 Feb 28;26(1):E114-E125. doi: 10.1532/hsf.5211. Raja SG, Uzzaman M, Garg S, Santhirakumaran G, Lee M, Soni MK, Khan H. Comparison of minimally invasive direct coronary artery bypass and drug-eluting stents for management of isolated left anterior descending artery disease: a systematic review and meta-analysis of 7,710 patients. Ann Cardiothorac Surg. 2018 Sep;7(5):567-576. doi: 10.21037/acs.2018.06.16.
Comment 7: A more robust discussion of the limitations and challenges of minimally invasive techniques would add depth and credibility.
Response 7: Thank you for your insightful feedback regarding the discussion of limitations and challenges associated with minimally invasive techniques. I fully recognize that addressing these aspects in depth enhances the credibility and balance of the review.
To incorporate this perspective, I have included a subsection titled Technical Challenges (Page 9 and 10) that provides a detailed discussion of the limitations and drawbacks of minimally invasive coronary surgery. This subsection explores factors such as the learning curve, operative times, potential for incomplete revascularization, and the need for hybrid approaches in complex cases.
Round 2
Reviewer 2 Report
Comments and Suggestions for Authors
With these corrections the paper is more precise
Author Response
Comment 1: With these corrections the paper is more precise.
Response: I am grateful to the learned reviewer for his/hers supportive comments.